



**Comparison of statistical and analytical hierarchy process methods on flood susceptibility**
**mapping: in a case study of Tana sub-basin in northwestern Ethiopia**
Azemeraw Wubalem*[1], Gashaw Tesfaw[1], Zerihun Dawit[1], Belete Getahun[1], Tamirat
Mekuria[1], Muralitharan Jothimani[1]
1. Department of Geology, College of Natural and Computational Sciences, University of Gondar,
Ethiopia**,** P.X.BOX 196
**Abstract**: The sub-basin of Lake Tana is one of the most flood-prone areas in northwestern
Ethiopia, which is affected by flood hazards. Flood susceptibility modeling in this area is essential
for hazard reduction purposes. For this, the analytical hierarchy process (AHP), bivariate, and
multivariate statistical methods were used. Using an intensive field survey, historical record, and
Google Earth Imagery, 1404 flood locations were determined which are classified into 70%
training datasets and 30% testing flood datasets using subset in the GIS tool. The statistical
relationship between the probability of flood occurrence and eleven flood-driving factors is
performed using the GIS tool. Then, the flood susceptibility map of the area is developed by
summing all weighted factors using a raster calculator and classified into very low, low, moderate,
high, and very high susceptibility classes using the natural breaks method. The results for the area
under the curve (AUC) are 99.1% for the frequency ratio model is better than 86.9% using AHP,
81.4% using the logistic regression model, and 78.2% using the information value model. Based
on the AUC values, the frequency ratio (FR) model is relatively better followed by the AHP model
for regional flood use planning, flood hazard mitigation, and prevention purposes.
*Keywords: flood, susceptibility, Geographic Information System (GIS),* **analytical hierarchy**
**process (AHP),** *frequency ratio, information value, logistic regression, Ethiopia*
**Introduction**
A flood is an overflow of water that submerges usually dry land. It can also occur in rivers or lakes
when the flow rate exceeds the capacity of rivers channel, particularly at the bends or meanders in
the waterway and backflow from the Lakes. Natural hazards, in particular flood, has been affecting
the world during rainy seasons. Even though Flood is one of the natural parts of the hydrological
cycle, it is increased in both frequency and magnitude from year to year. This is because of the
over change of climate and land degradation on the Earth due to the anthropogenic intervention.



The anthropogenic intervention on the Earth can reduce the water retention capacity of the
catchments because of the cleanup of forestation for a different purpose, which resulted in a high
rate of soil erosions. The Flood hazard has been causing damage to crops, infrastructures,
engineering structures, properties, and loss of human and animal lives worldwide including
Ethiopia. As reported by (Samanta et al., 201; Calil et al. 2015), the flood has resulted in a risk to
a human being (like loss of life, injury), properties (agricultural area, yield production, villages,
and buildings), communication systems (urban infrastructure, bridges, roads, and railway routes),
cultural heritage and ecosystems. (Zou et al., 2013; Calil et al., 2015) stated that more than 2000
deaths can occur within a single year and more than 75 million people have adversely affected
across the planet Earth by flood hazards.
Flood hazard is becoming one of the destructive natural hazards in Ethiopia followed by landslide
incidences and resulted in huge damages of properties, crops, farmlands, infrastructures, and loss
of life. For example, in the last two years, 2019-2020, flood hazard was displaced more than
500,000 people and damaged wide cultivated lands (more than 25, 000 ha cultivated lands),
damaged various engineering structures, destructed more than 35 houses, and loss of lives in
Amhara, Somali, Afar, SNNP, Dire Dwa, and Oromia regions of Ethiopia. The study area is one
of the severely affected areas by flooding which resulted in the loss of life, properties, destruction
of houses, roads, and more than 7, 000-hectare farmlands covered by various crops in the area.
These show that huge economic loss caused by flooding hazard that retards the sustainable
development of the economy of the country. Therefore, flood susceptibility mapping is one of the
most important elements for early warning systems or strategies to prevent and mitigate future
flood situation, which helps to reduce the negative results of flood hazard. Flood susceptibility
mapping can be also perceived as one of the ways of vulnerability assessment (Adger et al., 2006;
Jacinto et al. 2015).  In geohazard mapping, susceptibility/vulnerability, hazard and risk mapping
are the most important activities to understand, mapping and evaluating the spatiotemporal
condition and level of risk due to geohazards. These terms have different meanings but some
researchers use the terms interchangeably. Susceptibility refers to the probability of occurrence of
an event within particular type in a given location where as hazard refers the probability of
occurrence of an event within a particular type and magnitude in a given location within a reference
period. This means, susceptibility can be used to predict the spatial occurrence of an events, but



hazard can be used to predict the spatiotemporal occurrence of an events in a given terrain. The
term risk refers to the expected losses or damage by an events in a given regions which is the
products of susceptibility, hazard and elements at risk. Hence, the main objective of this study is
to prepare flood susceptibility map, this study only focus on flood susceptibility other than hazard
and risk. The flood susceptibility mapping has implementing using various methods by different
and numerous studies. These methods including qualitative (for example, analytical hierarchy
process (AHP), quantitative (machine learning, statistical), and hydrological based methods. The
hydrological methods are very simple and are based on a nonlinear concept and they are less
effective to model complex features like catchments (Sahoo et al., 2009). Nowadays, these
traditional methods have been replaced by automated and rule-based methods that are more
suitable for flood hazard mapping (Hostache et al., 2013). SWAT (Anjum et al., 2016) and
WetSpass (Nurmohamed et al., 2012) methods are examples of hydrological methods that are used
to produced spatial flood susceptibility models by integrated GIS and remote sensing tools.
Qualitative methods are an expert-driven approach, which required field experience specialists
(Rahmati et al., 2016; Dahri and Abida 2017). Rely on the experience and professional background
knowledge of experts and subjectivity is the drawback of these methods. An analytical hierarchy
process (AHP) is an example of a qualitative method used by many scholars to produce a flood
susceptibility model based on a multicriteria analysis framework (Karimi et al., 2018). Machine
learning techniques are advanced methods that used in flood susceptibility mapping, however, a
considerable processing time, the requirement of having high-performance computing systems
along with specific software, and strict selection criteria for input parameters make machine
learning methods less usable for a wide range of users (Ghalkhani et al., 2013; Tehrany et al. 2013).
Statistical methods are indirect susceptibility mapping methods widely or routinely used to
evaluate the correlation between flood driving factors and floods based on mathematical
expression (Bednarik et al., 2012; Chen and Wang, 2007; Pradhan et al., 2011; Regmi et al., 2014;
Wang et al., 2011). Statistical methods are imperative to utilize quick, understandable, and
accurate methods for flood susceptibility modeling. It has no specific requirements regarding input
data, software, and computer capacity. The statistical methods can be further divided into
multivariate and bivariate statistical methods, which are widely used throughout the world.  They
provide reliable results (Dai and Lepcha, 2002; Donati and Turrini, 2002; Luelseged and
Yamagishi, 2005; Duman et al., 2006; Sarkar et al., 2013; Meten et al., 2015; Chandak et al., 2016;



Kouhpeima et al., 2017; Wubalem and Meten, 2020;  Hong et al., 2020). The bivariate statistical
methods are used to evaluate the relationship between flood governing factors and past flooding.
Frequency ratio, certainty factor, information value, and weight of evidence are examples of
bivariate statistical methods, which are simple, easy, and produce reliable models. It also helps to
evaluate the effects of a flood at a factor class level that is impossible in data mining or multivariate
methods. However, it requires quality input data, past flood data, and lacking to evaluate the
relationship among flood governing factors. Multivariate statistical methods are used to examine
the relationship between three and above dependent and independent variables (Pham et al., 2016b;
Das, 2019; Duman et al., 2006; Kouhpeima et al, 2017; Luelseged and Yamagishi, 2005). Logistic
regression and discriminant analysis are examples of multivariate statistical methods used
frequently in flood susceptibility modeling and provide reliable results (Chen and Wang, 2007;
Das, 2019; Duman et al., 2006; Kouhpeima et al., 2017; Luelseged and Yamagishi, 2005; Meten
et al., 2015). However, it is incapable to examine the contribution of each factor class for flood
probability like data mining, unlike bivariate methods.
Many scholars have been employing both qualitative and quantitative methods for flood
susceptibility modeling, however, no clear and tangible agreements to select the best methods for
flood susceptibility modeling practice. Although the suitability of the model depend on various
constraints including physical parameters, data quality and availability, expert and technological
advancement, comparison among different natural hazard mapping methods is one of the solution
to select appropriate approaches. Hence, each methods has its own limitation, using different
approaches together for landslide or flood susceptibility mapping is very important to fill the gap
among the methods. For example, the logistic regression model can perform multivariate statistical
analysis between a dependent variable and a set of independent variables, but it is incapable to
analyze the impacts of internal classes of flood governing factors individually on flood occurrence.
This limitation can be solved using bivariate statistical methods, for example, frequency ratio and
information value statistical methods can be extracted the influence of each flood governing factor
class on flood occurrence, but it cannot consider the relationship between these flood governing
factors and flood occurrence. Therefore, a combination use of bivariate and multivariate statistical
methods are very essential to overcome the limitation of each methods. As a result, in the present
study, bivariate, multivariate and expert methods are employed to generate flood susceptibility





model in sub basin of Lake Tana and the performance of each methods has been evaluated using
receiver operating characteristics curve and area under the curve (AUC). Thus, based on the
concerns stated overhead, the main objective of this study is 1) to compare and evaluate the
performance of the frequency ratio, information value, logistic regression and analytical hierarchy
process methods to determine flood prone areas 2) to evaluate the relationship between flood
factors and flood probability as well as flood factor class and flood occurrence probability. The
nobility of this study lies on, 1) for the first time, the rigorous flood susceptibility methods like
statistical methods was conducted in the sub basin of Lake Tana to generate flood susceptibility
model 2) the comparison among the information value, frequency ratio, logistic regression and
analytical hierarchy process methods has not performed yet. This study will be determined
statistically significant methods for flood susceptibility modeling. The resulted map will be helped
the regional and local authorities and policy makers to mitigate flood hazards.

**Study Area**

The study area is located in Amhara Regional State of the sub-basin of Lake Tana basin in
northwestern Ethiopia, which is characterized, by wide flat to gently sloping plains and somehow
raged topography. Its elevation ranges from 1,774-4,037 m above mean sea level (Fig. 1). It is
bound between 330,000-410, 000 E and 1,280,000-1,350,000 N.  It is characterized by subtropical
to cool climatically zones with very high and prolonged rainfall in between Jun to October. The
study area is covered mainly three Districts including Fogera, Farta, and Libo Kemkem which is
frequently affected by flood hazards yearly during heavy and prolonged rainfall seasons. The study
area has many tributaries that drained to the two major rivers called Gumara and Ribb Rivers that
also drained to Lake Tana, which is the parts of the Abay basin. Agriculture is one of the most
dominant land use in the study area, which is performed more than two per year. The dominant
soil types in the study area including clay, loam, sandy loam, silty sand, fine to coarse sand, and
gravels sourced from volcanic rocks.


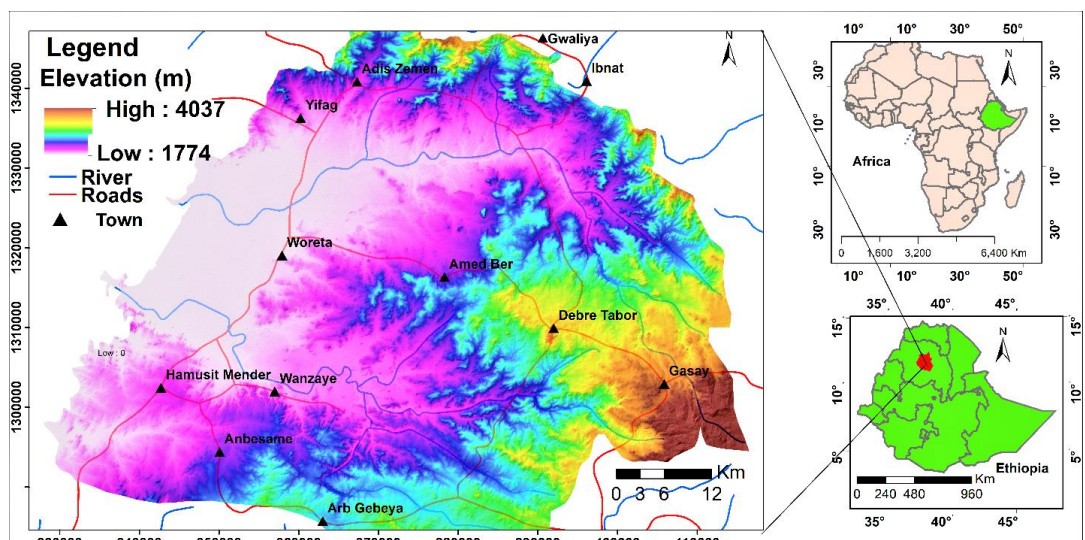

Figure 1 Location Map of the Study Area
**Data Used**
**Flood Inventory Map**
In flood susceptibility mapping, flood inventory mapping is one of the key element, which can be
prepared using various techniques like the aerial photograph or Google Earth Imagery
interpretation, field investigation, and evaluation of archived data coupled with GIS tool.
Evaluating and recognizing the correlation between flood driving factors and flood incidences is
required an accurate and precise flood inventory map (Pradhan et al., 2012; Tehrany and Jones,
2017; Mahyat et al., 2019). This flood inventory map can be prepared in map forms from the data
that can be collected from a satellite image or Google Earth Imagery interpretation, historical
records, and extensive field survey. In the present research work, 1404 most relevant flood
inventory data were collected from historical records, Google Earth Imagery interpretation, and
Extensive fieldwork (Fig. 2). In the literature, several suggestions are provided regarding the size
of flood samples to be used for modeling and model verification (Ohlmacher and Davis, 2003).
Therefore, based on a literature review, the flood inventory data was classified into 70%  (983)
flood for the training dataset and 30% (421) for testing datasets keeping their spatial distribution
using subset in ArcGIS 10.1 (Lee et al., 2012; Tehrany et al., 2013; Khosravi et al., 2016; Mahyat


et al., 2019) as shown in the figure. The same number of flood and non-flood points were chosen
for the logistic regression analysis.

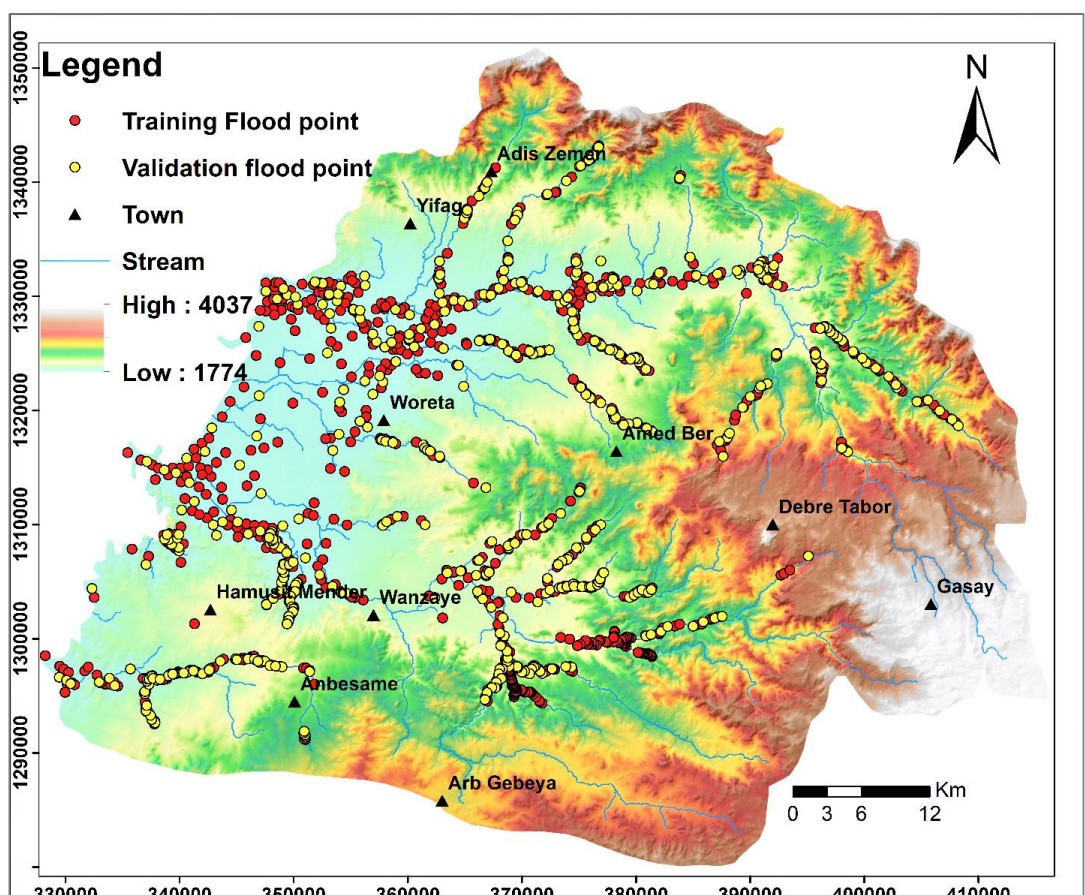


Figure 2 Flood location map
**Flood Driving Factors**
The selection of flood factors is one of the most crucial elements in flood susceptibility mapping,
which depend on physical and natural characteristics of the study area and data availability (Kia
et al., 2012; Liuzzo et al., 2019), however, no well-defined standards to select the most significant
flood driving factors. The factors that initiate the flood incidence in the study area are selected
based on the study area's environmental condition, data availability, logistic regression analysis,





and a literature review (Lee et al, 2012; Mahyat et al., 2019). The slope angle, slope curvature,
land use, soil texture, distance to stream/river, stream density, normalized vegetation index, flow
accumulation, groundwater depth, rainfall, and elevation have taken into account to examine the
spatial relationship between them and flood occurrence in the study area. These factors were
classified into subfactor classes using a natural break in ArcGIS to evaluate the effects of each
flood factor class for the case of frequency ratio and information vale methods. The flood factors,
which have derived from DEM, distance to stream (five classes), slope angle (five classes), flow
accumulation (five classes), stream density (five classes), elevation (five classes), and slope
curvature (three classes) maps were constructed from 12.5 m x 12.5 m resolution DEM (Fig. 3).
The soil map of the study area is prepared through digitization from a 1:50,000 textural soil map
of the Amhara Region, which has four classes (silty sand, sandy loam, clay, and loam). Land use
and NDVI maps of the study area were prepared from Sentinel 2 satellite image analysis using
ArcGIS with the help of high-resolution Google Earth image interpretation. The LULC has eight
classes including grazing land, agricultural land, barren land, residential/settlement, river zone
/water body, dense forest, moderate forest, and wetland (Fig. 3) whereas NDVI has five classes.
The rainfall and groundwater depth raster map was constructed using ArcGIS 10.1 from annual
mean rainfall and well data that are collected from Amhara Metrological Agency and Amhara
Water Well Drilling Enterprise, respectively. To determine the effects of each flood factor class
on flood occurrence, weight rating through flood factor raster combined with flood raster map is
important. For this purpose, all flood factor maps converted into a raster and reclassified with the
same pixel size (12.5 m x 12.5 m) and the same projection using the GIS tool. Then, the flood
inventory map is overlaid through a combination of spatial analysis tools under the local toolbox
with flood factor raster class to extracted flood pixels for each flood driving factor class. Then the
effects of each factor class were determined using the equation of frequency ratio, and information
value methods as summarized in Table 1.




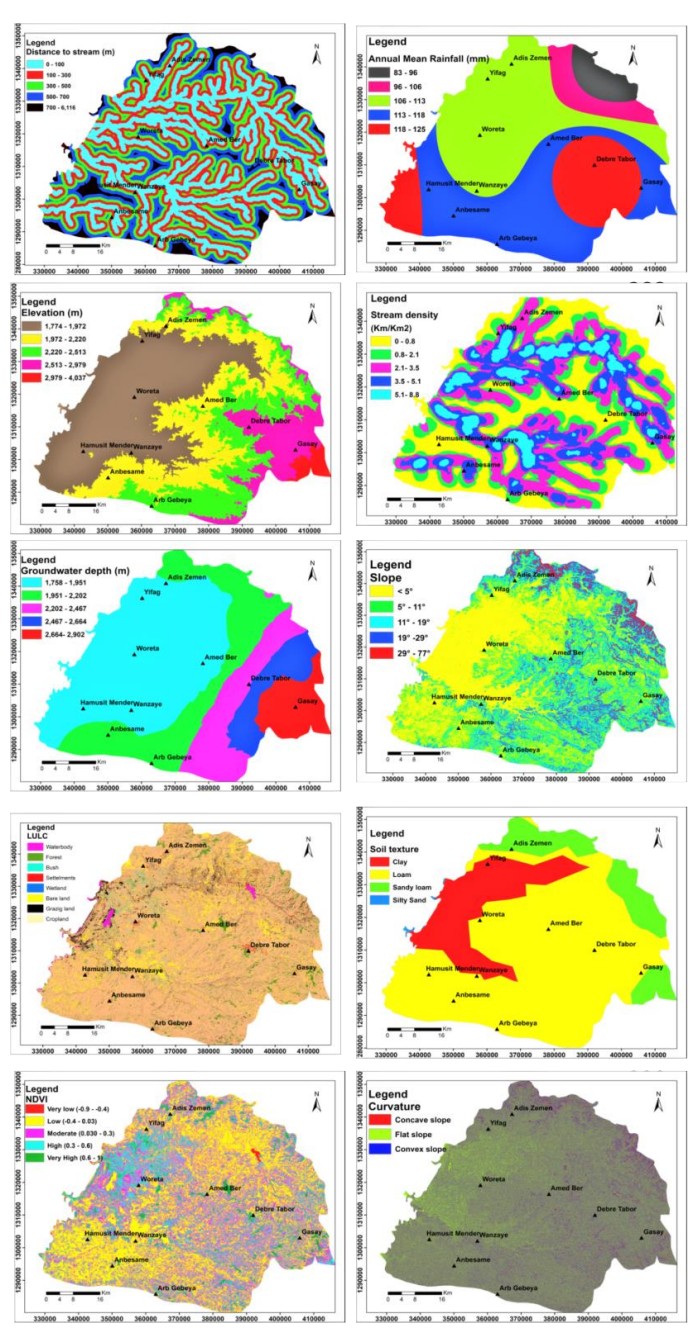


Figure 3 Flood governing factor maps




## Methodology

To achieve the goal of the present research work, various activities and steps are employed. These
are data collection, Flood inventory mapping, database creation for Flood factors, Flood
susceptibility modeling using frequency ratio, information value, logistic regression, and AHP
methods as well as model validation using the Receiver Operating Characteristics curve (ROC).
Moreover, appropriate data, including a topographic map, borehole data, Digital Elevation Model
(DEM) with 12.5 m resolution, historical flood events, soil type map, geological map, and
meteorological data were collected. These data were collected from the United States Geological
Survey (USGS), Amhara Water Well Drilling Enterprise (AWWDE), Field Survey, Google Earth
Imagery from the NASA, Ethiopian National Meteorological Agency, and the Geological Survey
of Ethiopia (GSE). The flood location of the study area identified using historical records, Google
Earth imagery analysis, and intensive field survey. This was classified into training and testing
flood datasets. The training flood datasets were used for model preparation, whereas the testing
flood datasets were used for model prediction accuracy evaluation. Based on the data availability,
local environmental conditions, data evaluation, literature, and local people interview, eleven
flood-driving factors were determined. The flood driving factor maps and flood inventory map
were prepared using ArcGIS 10.1.
Geodatabase building is one of the most fundamental elements in the flood susceptibility mapping.
Therefore, four databases were built for information value, logistic regression, frequency ratio, and
analytical hierarchy process (AHP) models. The frequency ratio, information value, and analytical
hierarchy process (AHP) database contain flood inventory and flood driving factors while the
logistic regression database contains flood and no flood points with eleven- weighted flood driving
factors. After the database was built, an evaluation of the relationship between flood and flood
factors as well as the determination of the statistical significance of each flood factor was the next
step in flood susceptibility mapping. Therefore, eleven flood factor maps reclassified into subclass
and overlaid with reclassified training flood datasets. Weight ratings for all flood factor classes
assigned statistically using Excel. These weighted maps rasterized-using lookup in spatial analyst.
After rasterized the factor maps, the flood susceptibility index maps were generated by the sum-
up of all raster maps using a raster calculator in Map Algebra. These maps (LSI) are classified into
a fivefold classification scheme: very low, low, moderate, high, and very high susceptibility classes
using natural breaks (Fig. 5, 6, 7, and 8). In the case of the logistic regression method, the study





area classified as training flood and non-flood points using GIS. Then, the weight of eleven factors
has been extracted to generate logistic regression coefficients of each flood factor in SPSS, and
finally, the flood susceptibility index of the area was generated using the logistic flood probability
equation (Eq. 8) and GIS tools (Fig. 3). Finally, the accuracy of the four models evaluated using
the prediction rate curve based on observed testing flood datasets (Fig. 9).
**Modeling Approaches**
**Information Value Model**
The information value method is one of the probabilistic methods of a bivariate statistical method,
which is used to envisage the correlation between floods and flood factor classes (Sakar et al.,
2006). The information values for each factor class determined through the combination of
reclassified flood raster to reclassified flood factor raster based on the presence of flood in a given
map unit. These values are important to define the role of each causal factor in classes for flood
occurrence. This can calculate as in Eq.1.
$$IV = \ln\left(\frac{\text{Conditional probability (CP)}}{\text{Prior probability (PP)}} = \frac{\frac{\text{Nfopix}}{\text{Ncpix}}}{\frac{\text{Ntfopix}}{\text{Ntcpix}}}\right) \qquad (1)$$
Where Conditional probability is the ratio of the pixel of a flood in class to the pixel of a class and
prior probability is the ratio of the total number of pixels of flood to the total number of pixels of
the study area. Nfopix is a flood pixel/area in a flood factor class. Ntfopix is the total area of a
flood in the entire study area. Ncpix is the area of the class in the study area and Ntcpix is the total
pixel area in the entire study area. When the IV > 0.1, the flood occurrence with the factor classes
have a high correlation, means it will have a high probability of flood occurrence however when
the IV < 0.1 or IV < 0, it is a low correlation between flood factors and flood occurrence which
indicate a low probability of flood occurrence. After calculated the information value for each
flood factor class using Microsoft excel and GIS, the information value for each factor class
assigned through the join in the ArcGIS tool. Then, the weighted flood factors rasterized using the
lookup tool in spatial analysis, and the flood susceptibility index (LSI) of the study area calculated
as in Eq. 2.



$$LSI = \sum_{i=1}^{n} IV_i X_i \qquad (2)$$

$LSI\ =\ IV * Slope\ raster\ +\ IV * drainage\ density + IV * groundwater\ depth + IV$
$* rainfall + IV * NDVI + IV * flow\ accumulation + IV * aspect\ raster$
$+\ IV * curvature\ raster\ + IV * soil\ raster\ +\ IV * Land\ use\ raster\ + IV$
$* Distance\ to\ stream\ raster$
Where LSI is the flood susceptibility index and IV is the information value of each factor class.
The higher value of LSI has indicated a higher probability of flood occurrence.
**Logistic Regression Model**
Logistic regression is one of the popular multivariate statistical analysis methods, which can be
used to establish a multivariate regression relationship between the dependent and independent
variables (Pradhan and Lee, 2010). Among other statistical methods, the logistic regression model
has been proven one of the most reliable approaches for flood susceptibility mapping to determine
the most flood influencing factors (Luelseged and Yamagishi, 2005; Chau and Chan, 2005; Lee
and Sanbath, 2006; Chen and Wang, 2007; Ricki and Graf, 2009]. This model is advantageous, as
it does not require normal distribution and it uses continuous or discrete variables. The difficulty
of using the logistic regression model lies in the sample size selection of dependent and
independent variables for flood susceptibility analysis. There are three ways of sampling flood and
non-flood points (Zhag et al., 2017). The first way is using all data from all the study areas.
However, this leads to an uneven proportion of non-flood and flood pixels, which incorporate a
large volume of data in the analysis. Using all flood pixels with equal non-flood pixels is the second
method, which also results in a less reliable output, but it can reduce sample size and sampling
bias. The third method uses an unequal or equal proportion of flood and non-flood pixels by
classifying flood into training and testing datasets.
In the present work, the floods of the study area were classified into training flood datasets (70%)
and as testing flood datasets (30%). In this study, the dependent data are a binary variable and are
made up of 0 and 1, which represent the absence and presence of floods, respectively.
Consequently, an equal number of non-flood sample points, whose dependent variable value is 0
where randomly selected from flood-free areas to represent the absence of floods using GIS. The





equal number of flood points and non-flood points were merged. Moreover, all the values of
independent variables containing flood and non-flood were extracted from the maps of each flood
governing factors using ArcGIS. Then, the logistic regression was conducted and coefficients were
calculated in the SPSS program. It can be expressed mathematically (Lee and Sambath, 2006;
Schicker and Moon, 2012) as:

$$P = \frac{1}{1 + e^{-z}} \qquad\qquad -----\qquad - (3)$$

Where P is the probability of flood occurrence that varies from zero to one. Z is the linear
combination of the predictors and varies from $-1 < z < 0$ for higher odds of non-flood occurrence
to $0 < z < 1$ for odds of higher flood occurrence. Z can be defined as:

$$Z = \beta_0 + \beta_1 X_1 + \beta_2 X_2 + \beta_3 X_3 \ldots \beta_n X_n BnXn ------------- (4)$$

Where $x_1, x_2, x_3 \ldots x_n$ are independent variables, Bo is the intercept of the slope of logistic regression
analysis, and $\beta_1, \beta_2, \beta_3 \ldots \beta_n$ are the coefficients of the logistic regression analysis.

**Frequency Ratio Model**

It is one of the bivariate probability methods, which is applicable to determine the correlation
between flood occurrence and flood causative factor classes. The frequency ratio is the ratio of
areas where the flood occurred in the areas to areas in which flood has not occurred. When the
ratio value is greater than one, it indicates the strong correlation between factor class and flood
occurrence in a given terrain, however, the ratio value less than one indicated that weak correlation
between flood occurrence and flood factors, which means a low probability of flood occurrence
(Lee and Talib, 2005). It can calculate using Eq. 5.

$$FR = \frac{a}{b} = \frac{\dfrac{\text{Nfopix}}{\text{Ntfopix}}}{\dfrac{\text{Ncpix}}{\text{Ntcpix}}} \qquad\qquad (5)$$

Where FR is frequency ratio, Nfopix is a flood pixel/area in a flood factor class, Ntfopix is the
total area of a flood in the entire study area (**a**), Ncpix is an area of the class in the study area and
Ntcpix is the total pixel area in the entire study area (**b**). In the present research work, the frequency
ratio for each causative factor class calculated using Eq.5, and the results summarized in Table 1.



After calculated the frequency ratio for each flood factor class using Microsoft Excel and GIS, the
frequency ratio value for each factor class assigned through the join in the ArcGIS tool. Then the
weighted flood factors rasterized using the lookup tool in spatial analysis. The flood susceptibility
index (LSI) of the study area was calculated by carefully summing up the weighted factor raster
maps using Eq. 6 by the raster calculator in Map Algebra of the spatial analysis tool. To get the
flood susceptibility index, the frequency ratio of each factor type or class is summed as in Eq. 6.
The flood susceptibility index indicated the degree of susceptibility of the area for flood
occurrence.
$$LSI = \sum_{i=1}^{n} FR_i X_i \qquad (6)$$

$LSI = FR * Slope\ raster + FR * drainage\ density + FR * groundwater\ depth + FR$
$* rainfall + FR * NDVI + FR * flow\ accumlation + FR * aspect\ raster$
$+ FR * curvature\ raster + FR * soil\ raster + FR * Land\ use\ raster$
$+ FR * Distance\ to\ stream\ raster$
Where LSI is the flood susceptibility index, n is the number of flood factors, $X_i$ is the flood factor
and $FR_i$ is the frequency ratio of each flood factor type or classes. After the flood susceptibility
index was calculated, the index values were classified into a different level of flood susceptibility
zones using natural breaks in the ArcGIS tool.  The higher the value of the flood susceptibility
index (LSI), the higher the probability of flood occurrence, but the lower the LSI indicates, the
lower the probability of flood occurrence.
Based on the natural break classification, the flood susceptibility map of the study area has five
classes such as very low, low, moderate, high, and very high landslide susceptibility class (Fig. 5).
**Analytical Hierarchy Process (AHP)**
The AHP is one of the qualitative methods used to determine the relationship between flood factor
class and flood occurrence. The AHP method is a structured tool that is used to analyze difficult
decisions based on the mathematics and psychology (Cho et al., 2015; Nguyen et al., 2015; Saaty,
2000; Zhang et al., 2016). To produce weighting factors, the pairwise comparison method was
used by considered Saaty's ranking scale (Luu et al., 2018; Saaty, 2008). The consistency of
calculated weight for each flood factor class was examined by the consistency ratio, which is


calculated by Eq.7 (Luu et al., 2018; Saaty, 2001). When the consistency ratio (CR) is less than
0.1, the weight of factor class that is calculated using the comparison matrix is consistent but if it
is greater than 0.1, the comparison matrix is inconsistent and it should be revised. After the weight
of each factor class was determined, the flood susceptibility map was produced as showed in Eq.9
(Rahmati et al., 2016c).
$$CR = \frac{CI}{RI} \qquad (7) \qquad CI = \frac{\lambda_{max} - n}{n} \qquad (8)$$
$$FSI = \sum_{i=1}^{n} W_i * X_n \qquad (9)$$
$LSI = W * Slope\ raster + W * drainage\ density + W * groundwater\ depth + W$
$\qquad\qquad * rainfall + W * NDVI + W * flow\ accumlation + W * aspect\ raster + W$
$\qquad\qquad * curvature\ raster + W * soil\ raster + W * Land\ use\ raster + W$
$\qquad\qquad * distance\ to\ stream\ raster$
Where CR is consistency ratio, CI is consistency index, RI is the average random consistency
index of the judgment matrix and $\lambda_{max}$ is the largest eigenvalue derived from the paired comparison
matrix and n is the number of flood factor, Wi is the weight of the flood factor, $X_n$ is the flood
factors and FSI is flooded susceptibility index.

## Result and Discussion

## Correlation of Flood Factors and Flood Incidence

### Frequency Ratio Results

The frequency ratio method is used to calculate FR for each subclass of every flood-driving factor,
which is the ratio of flood occurrence ratio to the area ratio. The result of the FR is summarized in
Table 1. The greater the value of FR indicates a strong correlation between flood factor class and
flood occurrence, a higher probability of flood occurrence when FR greater than unity (Table 1
and Fig. 4). As the results of the analysis designated in Table 1 and Fig. 4), the FR value for the
first slope class, 0° - 5° is greater than 1, is indicating a higher probability of flood occurrence
which has 96% of a flooded area in the slope classes. This finding is consistent with other studies
(e.g., Rahmati and Pourghasemi, 2017; Tehrany et al., 2014; Shafizadeh et al., 2018). However,
the slope gradient greater than 5° has less correlation with flood occurrence. This result confirmed
that the concepts as the slope gradient increase, the probability of flood occurrence in a given train
will be decreased. Because the steeper the slope gradient, the higher will be the rate of downslope
water velocity however the lower the water concentration as well as the infiltration of rainwater
into the ground. Nevertheless, when the slope gradient decreases, the potential for surface water
concentration and rainwater infiltration into the ground will increase it depends on the hydraulic
behavior of soil in that region. The higher concentration of surface water will have resulted in a
high probability of flood incidence.
Slope curvature is another flood factor, which has three classes including Convex, Concave, and
flat slope shapes. As the results of the correlation analysis of curvature class with flood inventory
indicated in Table 1, the flat class received a higher FR value, indicating a strong correlation with
flood occurrence. 56.1 % of the flooded area is fall in this class. This is because of the higher
potential of rainwater concentration and low infiltration of rainwater due to its flatness and the
existence of impermeable soil formation. Hence, this class is flat; the overflow of the water from
the riverbed is high in a class that is why the flat portion of the curvature class indicating higher
flood occurrence probability. This finding is confirmed with the other studies (Cao et al., 2016;
Chapi et al., 2017; Khosravi et al., 2016; Shafizadeh et al., 2018).
Table 1 indicated that the FR value for elevation class is decreased as the elevation of the region
is increased (Shafizadeh et al., 2018), indicating higher flood probability correlation with the first
class of 1, 774 – 1, 972 m which is 99 % of the flooded area fall in this region. As indicated in
Table 1, the relationship between elevation and the relative likelihood of flood occurrence is a
negative correlation at the elevation > 1,972 m, meaning the probability of flood occurrence is low
in elevated lands than low lands (Shafizadeh et al., 2018). This result is similar to the previous
studies of (Hong et al., 2016; Shafizadeh et al., 2018).
In the spatial prediction of flood-prone areas in a catchment, distance to the river is a critical factor
because floods occur due to the overflowing of water from the riverbanks (Chapi et al., 2017).
Therefore, the areas closer to the riverbeds demonstrate a rapid response to rainstorms and
flooding. As the results of the analysis shown in Table 1, the first four classes (0 -100 m, 100 –
300 m, 300 – 500 m, and 500 – 700 m) indicating a strong correlation with flood occurrence and
57.1 % of flooded area falls in these classes but the value of FR is decreased as the distance to the
river bed is increased. This result confirmed that the concepts, the closer to the riverbed, the higher
would be flood occurrence probability (Chapi et al., 2017; Hong et al., 2020;  Shafizadeh et al.,



2018). As the correlation analysis of flow accumulation with flood inventory results indicated in
Table 1, flow accumulation is one of the most important parameters in flood susceptibility mapping
(Pradhan, 2010). The higher value of FR for flow accumulation is indicating higher concentration
water and consequently higher flood occurrence probability. As Table 1 indicated, when the flow
accumulation increased, the FR value is increased in parallel. Land use and land cover are other
important parameters in flood susceptibility mapping which can be influenced by the
interrelationship between surface and groundwater, the amount of infiltration, surface water
concentration, and overland flow. As the result of land use and flood inventory correlation analysis
indicated in Table 1, River zone, barren land, grazing land, settlement, and moderate
vegetation/cropland have higher FR value, indicating higher flood occurrence probability. 37% of
flooded area falls in these land-use classes. Because the moderate vegetation/cropland favors
rainwater infiltration and hence the groundwater of this region is shallow, which enhanced the
overland flow of water that is why moderate vegetation class has received higher FR value. The
urban and grazing land have received higher FR value because of the impermeable nature of the
class and indicating higher flood occurrence probability correlation. This result is in line with the
work of (Shafizadeh et al., 2018). The NDVI is one of the important parameters for flood
susceptibility mapping, its value ranges from – 1 to 1. When the value is closer to one, the higher
vegetation cover but the closer to -1 implies the lower vegetation cover. Higher NDVI indicated
dense vegetation that can reduce and slow water flow (Turoglu and Dolek, 2011). This gives the
water time to infiltrate into the ground and resulting in a decrease in water volume and less
probability of flood occurrence. However, it depends on the hydraulic behavior of soil and the
depth of groundwater. In this study, the NDVI value ranges from – 1 to 1 which is from non-
vegetated to highly vegetated regions. As the vegetation density increased, the flood susceptibility
of a region will be decreased depending on the depth of groundwater and vegetation type. As the
results of NDVI with flood inventory correlation analysis indicated in Table 1, the first, third,
fourth, and fifth classes of the NDVI have received a higher value of FR and indicating higher
flood occurrence probability correlation.  This is because the groundwater depth of the study area
is shallow which can be increased overland flow water by reducing the rate of infiltration of
rainwater that is why the region shows higher flood occurrence correlation. 60.4% of the flooded
area falls in these classes. Table 1 shows, as a stream density increased, the value of FR is increased
in parallel and indicating high flood occurrence probability (Chapi et al., 2017; Shafizadeh et al.,





2018).  The stream density classes (3.5 – 5.1 m/km$^2$ and 5.1 – 8.8 m/km$^2$) have received a high
value of FR, indicating a strong correlation with flood occurrence and 61.5 % of flooded area falls
in these classes.
The amount of surface water concentration and rainwater infiltration rate mainly depends on the
hydraulic behavior of soils in the region. When the soil mass in a region is highly pervious, the
rate of water infiltration into the ground would be higher but the amount of surface water
concentration would be lower. This will enhance the non-flood incidence probability in a region.
However, this will be highly affected by the depth of groundwater. The results of flood inventory
with soil correlation analysis indicated in Table 1, silty sand and clay soil mass have received
higher value of FR compared to loam and sandy loam soil masses, indicating higher flood
incidence probability.  This is because of the impervious behavior of fine-grained soils. When the
grain size of soil mass increased, the percent of pore space in between soil grain will increase but
the pore space diameter will low. This leads to the blockage of flowing water inside the soil. These
types of soil will have a high water holding capacity. This again increased the overland flow of
water. This can be contributed to high flood incidence probability. 88% of the flooded area falls
in the silty sand and clay soil masses.  Table 1 indicated the shallow groundwater class has received
a high value of FR, indicating high flood incidence probability. 97.2 % of the flooded area falls in
very shallow groundwater depth. Even though rainfall is one of the most important flood driving
factors, its effect highly depends on the nature of the ground and the depth of the river channel. As
a result of rainfall with flood inventory analysis indicated in Table 1, the annual mean rainfall of
class (106 – 113 mm) has received a high value of FR, indicating high flood incidence probability.
This is because of the impervious hydraulic behavior of soil mass, low slope gradient, and shallow
groundwater depth. 68.5 % of the flooded area falls in the class (106 – 113 mm).




Figure 4 Statistical relationship between flood occurrence and flood driving factors



## Information value Results

ArcGIS 10.2 and Microsoft Excel were used to calculate the information value (IV) of each factor
classes to determine the statistical significance of each factor class for flood incidence probability.
The factor class, which received higher (positive) information value indicating higher flood
occurrence probability, but the factor class, which has received lower (negative) information value
indicating a negative or weak correlation with flood occurrence probability. For example, as the
result shown in Table 1, the distance to the stream of the first four classes indicating a positive
correlation with flood occurrence but the rest factor class of the distance to stream, show negative
correlations for flood occurrence probability. The slope class > 5°, elevation > 1, 972 m, the first
class of flow accumulation, distance to stream class > 700 m, the stream density classes $(0 – 0.8$
$Km^2$, $0.8 – 2.1 Km^2$, and $2.1 – 3.5 Km^2$), slope curvature (concave & convex slope), LULC (dense
forest, wetland, and agriculture land), the second and the third classes of NDVI, Soil texture (sandy
loam & loam), and groundwater depth > 1, 951 m did show negative statistical correlation with
flood occurrence probability (Table 1).
Table 1 Statistical analysis results of flood occurrence and flood factors using FR, and IV methods

| Slope Class | Class Pixel | % Class Pixel (b) | Flooded Area Pixel | % Flooded Area (a) | FR = a/b | Con_P | Prio_P | Con_P/Prior_P | IV = ln(Con_P/Prio_P) |
|---|---|---|---|---|---|---|---|---|---|
| < 5° | 11426722 | 45.18 | 507460 | 95.99 | 2.12 | 0.044 | 0.02 | 2.12 | 0.75 |
| 5° - 11° | 6780625 | 26.81 | 18543 | 3.51 | 0.13 | 0.003 | 0.02 | 0.13 | -2.03 |
| 11° - 19° | 4173258 | 16.50 | 2457 | 0.46 | 0.03 | 0.001 | 0.02 | 0.03 | -3.57 |
| 19° -29° | 2159660 | 8.54 | 207 | 0.04 | 0.00 | 0.000 | 0.02 | 0.00 | -5.38 |
| 29° - 77° | 750796 | 2.97 | 8 | 0.00 | 0.00 | 0.000 | 0.02 | 0.00 | -7.58 |
| **Elevation** | | | | | | | | | |
| Class (m) | Class Pixel | % Class Pixel (b) | Flooded Area Pixel | % Flooded Area (a) | FR = a/b | Con_P | Prio_P | Con_P/Prior_P | IV = ln(Con_P/Prio_P) |
| 1,774 - 1972 | 10237743 | 40.48 | 523039 | 98.93 | 2.44 | 0.051 | 0.02 | 2.44 | 0.89 |
| 1, 972 - 2, 220 | 6383841 | 25.24 | 5292 | 1.00 | 0.04 | 0.001 | 0.02 | 0.04 | -3.23 |
| 2,220 - 2,513 | 5148369 | 20.36 | 344 | 0.07 | 0.00 | 0.000 | 0.02 | 0.00 | -5.75 |
| 2,513 - 2,979 | 3038070 | 12.01 | 0 | 0.00 | 0.00 | 0.000 | 0.02 | 0.00 | |
| 2,979 - 4,037 | 483038 | 1.91 | 0 | 0.00 | 0.00 | 0.000 | 0.02 | 0.00 | |
| **Flow Accumulation** | | | | | | | | | |
| Class | Class Pixel | % Class Pixel (b) | Flooded Area Pixel | % Flooded Area (a) | FR = a/b | Con_P | Prio_P | Con_P/Prior_P | IV = ln(Con_P/Prio_P) |
| Very low | 25250270 | 99.84 | 524941 | 99.29 | 0.99 | 0.021 | 0.02 | 0.99 | -0.01 |
| Low | 25502 | 0.10 | 1653 | 0.31 | 3.10 | 0.065 | 0.02 | 3.10 | 1.13 |
| Moderate | 8076 | 0.03 | 1037 | 0.20 | 6.14 | 0.128 | 0.02 | 6.14 | 1.82 |
| High | 3257 | 0.01 | 532 | 0.10 | 7.81 | 0.163 | 0.02 | 7.81 | 2.06 |
| Very high | 3956 | 0.02 | 512 | 0.10 | 6.19 | 0.129 | 0.02 | 6.19 | 1.82 |
| **Distance to Stream** | | | | | | | | | |
| Class (m) | Class Pixel | % Class Pixel (b) | Flooded Area Pixel | % Flooded Area (a) | FR = a/b | Con_P | Prio_P | Con_P/Prior_P | IV = ln(Con_P/Prio_P) |
| 0 - 100 | 1310596 | 5.18 | 75517 | 14.28 | 2.76 | 0.058 | 0.02 | 2.76 | 1.01 |




| 100 - 300 | 2399920 | 9.49 | 99494 | 18.82 | 1.98 | 0.041 | 0.02 | 1.98 | 0.68 |
| 300 - 500 | 2288224 | 9.05 | 73168 | 13.84 | 1.53 | 0.032 | 0.02 | 1.53 | 0.43 |
| 500 - 700 | 2153831 | 8.52 | 53693 | 10.16 | 1.19 | 0.025 | 0.02 | 1.19 | 0.18 |
| 700 -6,116.5 | 17138490 | 67.77 | 226803 | 42.90 | 0.63 | 0.013 | 0.02 | 0.63 | -0.46 |

**Stream Density**

| Class (Km2) | Class Pixel | % Class Pixel (b) | Flooded Area Pixel | % Flooded Area (a) | FR = a/b | Con_P | Prio_P | Con_P/Prior_P | IV = ln(Con_P/Prio_P) |
|---|---|---|---|---|---|---|---|---|---|
| 0 - 0.8 | 6882039 | 27.92 | 46291 | 8.76 | 0.31 | 0.007 | 0.02 | 0.31 | -1.16 |
| 0.8 - 2.1 | 4983095 | 20.21 | 58174 | 11.00 | 0.54 | 0.012 | 0.02 | 0.54 | -0.61 |
| 2.1 - 3.5 | 6317902 | 25.63 | 99289 | 18.78 | 0.73 | 0.016 | 0.02 | 0.73 | -0.31 |
| 3.5 - 5.1 | 4350662 | 17.65 | 174722 | 33.05 | 1.87 | 0.040 | 0.02 | 1.87 | 0.63 |
| 5.1 - 8.8 | 2118013 | 8.59 | 150199 | 28.41 | 3.31 | 0.071 | 0.02 | 3.31 | 1.20 |

**Slope Curvature**

| Class | Class Pixel | % Class Pixel (b) | Flooded Area Pixel | % Flooded Area (a) | FR = a/b | Con_P | Prio_P | Con_P/Prior_P | IV = ln(Con_P/Prio_P) |
|---|---|---|---|---|---|---|---|---|---|
| Concave | 4388463 | 17.35 | 71032 | 13.44 | 0.77 | 0.016 | 0.02 | 0.77 | -0.26 |
| Flat slope | 11840022 | 46.82 | 296510 | 56.09 | 1.20 | 0.025 | 0.02 | 1.20 | 0.18 |
| Convex slope | 9062576 | 35.83 | 161133 | 30.48 | 0.85 | 0.018 | 0.02 | 0.85 | -0.16 |

**LULC**

| Class name | Class Pixel | % Class Pixel (b) | Flooded Area Pixel | % Flooded Area (a) | FR = a/b | Con_P | Prio_P | Con_P/Prior_P | IV = ln(Con_P/Prio_P) |
|---|---|---|---|---|---|---|---|---|---|
| Waterbody | 170378 | 0.67 | 53875 | 10.19 | 15.13 | 0.316 | 0.02 | 15.13 | 2.72 |
| Dense forest | 1584350 | 6.27 | 25202 | 4.77 | 0.76 | 0.016 | 0.02 | 0.76 | -0.27 |
| Moderate forest | 185078 | 0.73 | 8200 | 1.55 | 2.12 | 0.044 | 0.02 | 2.12 | 0.75 |
| Settlements | 291928 | 1.15 | 9189 | 1.74 | 1.51 | 0.031 | 0.02 | 1.51 | 0.41 |
| Wetland | 72446 | 0.29 | 812 | 0.15 | 0.54 | 0.011 | 0.02 | 0.54 | -0.62 |
| Bare land | 2288296 | 9.05 | 52119 | 9.86 | 1.09 | 0.023 | 0.02 | 1.09 | 0.09 |
| Grazing land | 1017397 | 4.02 | 71962 | 13.61 | 3.38 | 0.071 | 0.02 | 3.38 | 1.22 |
| Agricultural land | 19678779 | 77.82 | 307316 | 58.13 | 0.75 | 0.016 | 0.02 | 0.75 | -0.29 |

**NDVI**

| Class | Class Pixel | % Class Pixel (b) | Flooded Area Pixel | % Flooded Area (a) | FR = a/b | Con_P | Prio_P | Con_P/Prior_P | IV = ln(Con_P/Prio_P) |
|---|---|---|---|---|---|---|---|---|---|
| Very Low | 101398 | 0.26 | 3352 | 0.63 | 2.47 | 0.033 | 0.01 | 2.47 | 0.90 |
| Low | 18762295 | 47.48 | 209315 | 39.59 | 0.83 | 0.011 | 0.01 | 0.83 | -0.18 |
| Moderate | 11571866 | 29.28 | 165084 | 31.23 | 1.07 | 0.014 | 0.01 | 1.07 | 0.06 |
| High | 6389220 | 16.17 | 100817 | 19.07 | 1.18 | 0.016 | 0.01 | 1.18 | 0.17 |
| Very High | 2692708 | 6.81 | 50107 | 9.48 | 1.39 | 0.019 | 0.01 | 1.39 | 0.33 |

**Soil Texture**

| Class | Class Pixel | % Class Pixel (b) | Flooded Area Pixel | % Flooded Area (a) | FR = a/b | Con_P | Prio_P | Con_P/Prior_P | IV = ln(Con_P/Prio_P) |
|---|---|---|---|---|---|---|---|---|---|
| Loam | 17424445 | 68.91 | 61780 | 11.69 | 0.17 | 0.004 | 0.02 | 0.17 | -1.77 |
| Silty Sand | 33442 | 0.13 | 2502 | 0.47 | 3.58 | 0.075 | 0.02 | 3.58 | 1.27 |
| Clay | 4514511 | 17.85 | 462543 | 87.50 | 4.90 | 0.102 | 0.02 | 4.90 | 1.59 |
| Sandy loam | 3313428 | 13.10 | 1809 | 0.34 | 0.03 | 0.001 | 0.02 | 0.03 | -3.65 |

**Groundwater**

| Class | Class Pixel | % Class Pixel (b) | Flooded Area Pixel | % Flooded Area (a) | FR = a/b | Con_P | Prio_P | Con_P/Prior_P | IV = ln(Con_P/Prio_P) |
|---|---|---|---|---|---|---|---|---|---|
| 1,750 -1, 951 | 11643964 | 46.04 | 514021 | 97.23 | 2.11 | 0.044 | 0.02 | 2.11 | 0.75 |
| 1,951 - 2, 202 | 6130330 | 24.24 | 9137 | 1.73 | 0.07 | 0.001 | 0.02 | 0.07 | -2.64 |
| 2, 202 - 2, 467 | 3367218 | 13.31 | 3792 | 0.72 | 0.05 | 0.001 | 0.02 | 0.05 | -2.92 |
| 2, 467 - 2, 664 | 2064256 | 8.16 | 1725 | 0.33 | 0.04 | 0.001 | 0.02 | 0.04 | -3.22 |
| 2, 664 - 2, 902 | 2085293 | 8.25 | | 0.00 | 0.00 | 0.000 | 0.02 | 0.00 | |


| Rainfall | | | | | | | | | |
|---|---|---|---|---|---|---|---|---|---|
| Class | Class Pixel | % Class Pixel (b) | Flooded Area Pixel | % Flooded Area (a) | FR = a/b | Con_P | Prio_P | Con_P/Prior_P | IV = ln(Con_P/Prio_P) |
| 83 - 96 | 1207382 | 4.77 | 1763 | 0.33 | 0.07 | 0.001 | 0.02 | 0.07 | -2.66 |
| 96 - 106 | 1641787 | 6.49 | 6480 | 1.23 | 0.19 | 0.004 | 0.02 | 0.19 | -1.67 |
| 106 - 113 | 8856030 | 35.02 | 362318 | 68.53 | 1.96 | 0.041 | 0.02 | 1.96 | 0.67 |
| 113 - 118 | 8706231 | 34.42 | 139032 | 26.30 | 0.76 | 0.016 | 0.02 | 0.76 | -0.27 |
| 118 - 125 | 4879631 | 19.29 | 19082 | 3.61 | 0.19 | 0.004 | 0.02 | 0.19 | -1.68 |
| IV is information value, FR is frequency ratio, a is flooded area in a factor class, b is an area of factor class, Con_P is conditional probability and Prio_P is the prior probability | | | | | | | | | |


## Logistic Regression Results

Hence, sets of independent variables are so sensitive for collinearity (interrelatedness of independent variable) which can be checked using Tolerance (TOL) and variance inflation factor index (VIF), Multicollinearity test was applied using SPSS software before logistic regression analysis. When the Tolerance (TOL) < 0.2 and VIF > 5, the given independent variable have multicollinearity. As a result of the multicollinearity test indicated in Table 2, no independent variables that were used in flood susceptibility analysis showed any multicollinearity. Using logistic regression analysis in SPSS, the logistic regression coefficient for all flood-driving factors was determined. Similar to the information value method, the positive logistic regression coefficients indicating a positive association with flood occurrence probability but the negative logistic regression coefficients indicating a negative correlation of flood factors with flood occurrence probability. As the result of logistic regression analysis indicated in Table 2, Stream density, NDVI, Rainfall, and Curvature have received negative logistic regression coefficients but the remain factors that have received positive logistic regression coefficients, indicating the flood factors have positively associated with flood occurrence probability.

Table 2 logistic coefficients of flood factors and multicollinearity statistics

| Factors | LR Coefficients(β) | Collinearity Statistics | |
|---|---|---|---|
| | | Tolerance (TOL) | Variance inflation factor index (VIF) |
| Curvature | -0.04 | 0.983 | 1.017 |
| Elevation | 0.804 | 0.441 | 2.267 |
| Flow Accumulation | 0.222 | 0.957 | 1.045 |
| Groundwater Depth | 0.006 | 0.485 | 2.062 |
| LULC | 0.159 | 0.947 | 1.056 |
| NDVI | -1.198 | 0.925 | 1.081 |
| Rainfall | -0.148 | 0.652 | 1.534 |
| Slope | 0.769 | 0.608 | 1.644 |
| Soil Texture | 0.106 | 0.58 | 1.724 |



| | | | |
|---|---|---|---|
| Distance to Stream | 1.73 | 0.61 | 1.641 |
| Stream Density | -0.095 | 0.65 | 1.538 |
| Constant | -4.383 | | |


## AHP Pairwise Comparison Matrix Results

After reclassifying and ranking of the eleven-flood factor thematic raster into subclasses, the
pairwise comparison was performed for 5 x 5, 8 x 8, 4 x 4 and 3 x 3 matrixes using AHP calculator
(Table 3), where the diagonal element is equal to 1.  As indicated in Table 3, the significance of
sub-criteria for each factor has shown in the row of the pairwise comparison matrix. The first row
in the Table 3 illustrates the significance of the first slope angle compared to the other slope angle
classes. For instance, the first slope angle class $(0° – 5°)$ is significantly more important than the
other slope classes, which are placed in the column for flood probability and assigned 9. However,
for the last classes of the slope angle at the row has less significant for flood probability and
assigned the reciprocal values of the pairwise comparison (E.g. 1/9 for the last slope class, 29° -
77°). The details for all parameters weight rating have summarized in Table 3 and the consistency
of the factor class weight was evaluated using the consistency ratio (CR). When CR < 0.1, the
weights' consistency is affirmed. As indicated in Table 3, the CR value for all factor classes is less
than 0.1 and indicated no weights' inconsistency. Based on the results of the pairwise comparison
analysis, as the slope angle, elevation, and groundwater depth increased, the flood probability will
be decreased and the vise verse. Similarly, as the distance to Riverbed increased, the flood
probability will be decreased. Concerning the other parameters, as the stream density, rainfall and
flow accumulation increased, the flood probability will be increased (Table 3). The flood
occurrence probability and its impact also depend on the hydraulic behavior of soil regard to the
other parameters. If the permeability of soil is high, the flood probability will low. This depends
on the grain size and diameters of pore space between soil particles. Therefore, the clay soil has
low permeability than high water holding capacity. This is the case why the clay soil has received
high value (9) in the pairwise comparison matrix (Table 3). In the study area, Settlement, bare
land, agricultural land, grazing land, water body, and wetland have a high contribution to flood
occurrence respectively compared to the forested regions.



Table 3 Pairwise comparison matrix and weight of flood factor classes

| Factors | Sub Factor Class(i) | Sub Factor Class (j) | | | | | |
|---|---|---|---|---|---|---|---|
| **Slope** | Class | 0° - 0.5° | 0.85°- 11 ° | 11° - 19° | 19° - 29 ° | 29° - 77° | **W** |
| | 0° - 5° | 1 | 2 | 5 | 7 | 9 | **0.509** |
| | 5°- 11 ° | 0.5 | 1 | 2 | 3 | 4 | **0.229** |
| | 11° - 19° | 0.2 | 0.5 | 1 | 2 | 5 | **0.143** |
| | 19° - 29 ° | 0.14 | 0.33 | 0.5 | 1 | 2 | **0.075** |
| | 29° - 77° | 0.11 | 0.25 | 0.2 | 0.5 | 1 | **0.044** |
| | **Consistency Ratio CR = 5.9%** | | | | | | |
| **Elevation** | Class (m) | 1,774 - 1972 | 1, 972 - 2, 220 | 2,220 - 2,513 | 2,513 - 2,979 | 2,979 - 4,037 | **W** |
| | 1,774 - 1972 | 1 | 3 | 6 | 7 | 9 | **0.543** |
| | 1, 972 - 2, 220 | 0.33 | 1 | 3 | 3 | 4 | **0.222** |
| | 2,220 - 2,513 | 0.17 | 0.33 | 1 | 2 | 5 | **0.123** |
| | 2,513 - 2,979 | 0.14 | 0.33 | 0.5 | 1 | 2 | **0.071** |
| | 2,979 - 4,037 | 0.11 | 0.25 | 0.2 | 0.5 | 1 | **0.042** |
| | **Consistency Ratio CR = 1.4%** | | | | | | |
| **Flow** | Class | Very low | Low | Moderate | High | Very high | **W** |
| | Very low | 1 | 0.33 | 0.11 | 0.11 | 0.11 | **0.031** |
| | Low | 3 | 1 | 0.33 | 0.33 | 0.33 | **0.092** |
| | Moderate | 9 | 3 | 1 | 0.33 | 0.33 | **0.186** |
| | High | 9 | 3 | 3 | 1 | 1 | **0.346** |
| | Very high | 9 | 3 | 3 | 1 | 1 | **0.346** |
| | **Consistency Ratio CR = 4.4%** | | | | | | |
| **Distance Stream** | Class (m) | 0 - 100 | 100 - 300 | 300 - 500 | 500 - 700 | 700 -6,116.5 | **W** |
| | 0 - 100 | 1 | 3 | 5 | 9 | 9 | **0.529** |
| | 100 – 300 | 0.33 | 1 | 3 | 3 | 5 | **0.229** |
| | 300 – 500 | 0.2 | 0.33 | 1 | 3 | 7 | **0.147** |
| | 500 – 700 | 0.11 | 0.33 | 0.33 | 1 | 1 | **0.053** |
| | 700 -6,116.5 | 0.11 | 0.2 | 0.14 | 1 | 1 | **0.042** |
| | **Consistency Ratio CR = 6.5%** | | | | | | |
| **Stream Density** | Class (Km2) | 0 - 0.8 | 0.8 - 2.1 | 2.1 - 3.5 | 3.5 - 5.1 | 5.1 - 8.8 | **W** |
| | 0 - 0.8 | 1 | 0.33 | 0.2 | 0.14 | 0.11 | **0.033** |
| | 0.8 - 2.1 | 3 | 1 | 0.33 | 0.2 | 0.14 | **0.064** |
| | 2.1 - 3.5 | 5 | 3 | 1 | 0.33 | 0.14 | **0.124** |
| | 3.5 - 5.1 | 7 | 5 | 3 | 1 | 1 | **0.324** |
| | 5.1 - 8.8 | 9 | 7 | 7 | 1 | 1 | **0.455** |
| | **Consistency Ratio CR = 5.9%** | | | | | | |
| **NDVI** | Class | Very Low | Low | Moderate | High | Very High | **W** |
| | Very Low | 1 | 2 | 5 | 9 | 9 | **0.489** |
| | Low | 0.5 | 1 | 3 | 5 | 7 | **0.282** |
| | Moderate | 0.2 | 0.33 | 1 | 3 | 7 | **0.144** |
| | High | 0.11 | 0.2 | 0.33 | 1 | 1 | **0.047** |
| | Very High | 0.11 | 0.14 | 0.14 | 1 | 1 | **0.039** |
| | **Consistency Ratio CR = 4.4%** | | | | | | |
| **Rainfall** | Class | 83 - 96 | 96 - 106 | 106 - 113 | 113 - 118 | 118 - 125 | **W** |
| | 83 – 96 | 1 | 0.5 | 0.2 | 0.11 | 0.11 | **0.033** |
| | 96 – 106 | 2 | 1 | 0.33 | 0.2 | 0.14 | **0.057** |
| | 106 – 113 | 5 | 3 | 1 | 0.33 | 0.14 | **0.123** |
| | 113 – 118 | 9 | 5 | 3 | 1 | 1 | **0.335** |
| | 118 – 125 | 9 | 7 | 7 | 1 | 1 | **0.452** |
| | **Consistency Ratio CR = 4.4%** | | | | | | |
| **GW** | Class (m) | 1,750 -1, 951 | 1,951 - 2, 202 | 2, 202 - 2, 467 | 2, 467 - 2, 664 | 2, 664 - 2, 902 | **W** |
| | 1,750 -1, 951 | 1 | 3 | 5 | 9 | 9 | **0.522** |
| | 1,951 - 2, 202 | 0.33 | 1 | 3 | 5 | 7 | **0.256** |





| | 2, 202 - 2, 467 | 0.2 | 0.33 | 1 | 3 | 7 | **0.139** |
|---|---|---|---|---|---|---|---|
| | 2, 467 - 2, 664 | 0.11 | 0.2 | 0.33 | 1 | 1 | **0.046** |
| | 2, 664 - 2, 902 | 0.11 | 0.14 | 0.14 | 1 | 1 | **0.038** |

**Consistency Ratio CR** = 5.3%

| | Class | Waterbody | Dense Forest | Moderate Forest | Settlement | Wetland | Bare land | Grassing land | Agricultural land | W |
|---|---|---|---|---|---|---|---|---|---|---|
| | Waterbody | 1 | 4 | 4 | 0.33 | 2 | 0.5 | 0.5 | 0.33 | **0.091** |
| | Dense Forest | 0.25 | 1 | 0.5 | 0.2 | 0.14 | 0.11 | 0.11 | 0.11 | **0.021** |
| | Moderate Forest | 0.25 | 2 | 1 | 0.2 | 0.33 | 0.11 | 0.11 | 0.11 | **0.027** |
| **LULC** | Settlement | 3 | 5 | 5 | 1 | 3 | 1 | 2 | 2 | **0.225** |
| | Wetland | 0.5 | 7 | 3 | 0.33 | 1 | 0.5 | 0.5 | 0.33 | **0.08** |
| | Bare land | 2 | 9 | 9 | 1 | 2 | 1 | 2 | 1 | **0.204** |
| | Grassing land | 2 | 9 | 9 | 0.5 | 2 | 0.5 | 1 | 1 | **0.16** |
| | Agricultural land | 3 | 9 | 9 | 0.5 | 3 | 1 | 1 | 1 | **0.192** |

**Consistency Ratio CR** = 3.9%

| | Class | Loam | Silty Sand | Clay | Sandy loam | W |
|---|---|---|---|---|---|---|
| | Loam | 1 | 3 | 0.11 | 4 | **0.148** |
| **Soil** | Silty Sand | 0.33 | 1 | 0.11 | 2 | **0.07** |
| | Clay | 9 | 9 | 1 | 9 | **0.735** |
| | Sandy loam | 0.25 | 0.5 | 0.11 | 1 | **0.047** |

**Consistency Ratio CR** = 8.9%

| | Class | Concave | Flat slope | Convex slope | W |
|---|---|---|---|---|---|
| | Concave | 1 | 0.5 | 0.5 | **0.196** |
| **Curvature** | Flat slope | 2 | 1 | 2 | **0.493** |
| | Convex slope | 2 | 0.5 | 1 | **0.311** |

**Consistency Ratio CR** = 5.6%


## Flood Susceptibility Model

## Frequency Ratio Flood Susceptibility model

After weight rating for each flood driving factor classes using FR, each flood-driving factor was converted into raster using lookup in spatial analysis option under ArcGIS 10.2 software. The flood Susceptibility index of the study area is generated by sum up all raster maps carefully using the raster calculator in spatial analysis. The flood susceptibility index (Fig. 5) was reclassified into five classes (Very low, low, moderate, high, and very high) using the natural break method in ArcGIS as shown in Eq. 6. As a result, shown in Table 4, high and very high flood susceptibility classes have covered 19.8 % and 20.7 % of the study area, respectively. However, the remaining, 14.1 %, 23.6 %, and 21.7 % of the study area covered by very low, low, and moderate flood susceptibility areas. The high and very high flood susceptibility classes in the study area fell closer to the Ribb River, Gumara River, Ribb dam, and other streams as well as flat and impervious soil regions. However, the low and very low regions fell in the steep slope gradient and deep groundwater depth as well as densely forested and previous regions.




*FSI = FR\*Slope raster + FR\*Stream density raster + FR\*Slope curvature raster + FR\*Soil*
*Texture raster + FR\*Land use raster + FR\*Distance to stream raster +FR\*Flow Accumulation*
*+ FR\*Groundwater depth raster + FR\* Elevation raster + FR\*NDVI raster + FR\*Rainfall raster.*
**Information Value Flood Susceptibility Model**
Similar to the frequency ratio method, the flood susceptibility index generated using the
information value method (Fig. 6) was reclassified into five classes (Very low, low, moderate,
high, and very high) using the natural break method in ArcGIS as shown in Eq. 2. As a result,
shown in Table 4, high and very high flood susceptibility classes have covered 20.3 % and 20.2 %
of the study area, respectively. However, the remaining, 13.1 %, 23.9 %, and 22.5 % of the study
area covered by very low, low, and moderate flood susceptibility areas.
*FSI = IV\*Slope raster + IV\*Stream density raster + IV\*Slope curvature raster + IV\*Soil Texture*
*raster + IV\*Land use raster + IV\*Distance to stream raster +IV\*Flow Accumulation +*
*IV\*Groundwater depth raster + IV\* Elevation raster + IV\*NDVI raster + IV\*Rainfall raster*
**Logistic Regression Flood Susceptibility Model**
In the logistic regression method, logistic regression coefficients for individual factor was
determined using SPSS. The linear combination of LR constant and factor products with LR
coefficients is called Z, which is calculated as shown Eq. 4. The value of Z enters into Eq. 3 and
the flood probability index (P) was generated. The value of P is range from 0 – 1 and the closer
the value to one is indicating the higher flood susceptibility region. Similar to the frequency ratio
and information value methods, the flood susceptibility index generated using the logistic
regression method (Fig. 7) was reclassified into five classes (Very low, low, moderate, high, and
very high) using the natural break method in ArcGIS as shown in Eq. 3. As a result, shown in
Table 4, high and very high flood susceptibility classes have covered 13.2 % and 9.3 % of the
study area, respectively. However, the remaining, 54.3 %, 11.2 %, and 12.1 % of the study area
covered by very low, low, and moderate flood susceptibility area.
*Z =-4.38+ 0.769\*Slope raster + -0.095 \*Stream density raster + -0.040\*Slope curvature raster*
*+ 0.106\*Soil Texture raster + 0.159\*Land use raster + 1.73\*Distance to stream raster*
*+0.222\*Flow Accumulation + 0.006\*Groundwater depth raster + 0.804\* Elevation raster + -*
*1.198\*NDVI raster + -0.148\*Rainfall raster*



## Analytical Hieracky Process Flood Susceptibility Model


Similar to the frequency ratio and information value methods, the flood susceptibility index
generated using the analytical hieracky process method (Fig. 8) was reclassified into five classes
(Very low, low, moderate, high, and very high) using the natural break method in ArcGIS as shown
in Eq. 9. As a result, shown in Table 4, high and very high flood susceptibility classes have covered
19.8% and 10.2 % of the study area, respectively. However, the remaining, 19.7%, 24.8%, and
25.6% of the study area covered by very low, low, and moderate flood susceptibility areas.
$$LSI = W * Slope\ raster + W * drainage\ density + W * groundwater\ depth + W$$
$$\qquad * rainfall + W * NDVI + W * flow\ accumulation + W * aspect\ raster + W$$
$$\qquad * curvature\ raster + W * soil\ raster + W * Land\ use\ raster + W$$
$$\qquad * distance\ to\ stream\ raster$$
Table 4 Statistical model summary of FR, LR, IV, and AHP methods

| IVFSI | Class | IVFSP | % FSM | VFP | % VF | LRFSI | LRFSP | % FSM | VFP | % VF |
|---|---|---|---|---|---|---|---|---|---|---|
| -25 - -15.1 | Very low | 3226367 | 13.1 | 13 | 0.01 | 0 - 0.1 | 13381271 | 54.3 | 627 | 0.34 |
| -15.1 - -10 | Low | 5901361 | 23.9 | 472 | 0.26 | 0.1 - 0.3 | 2756345 | 11.2 | 4470 | 2.46 |
| - 10 - -5 | Moderate | 5535540 | 22.5 | 4816 | 2.65 | 0.3 - 0.5 | 2972834 | 12.1 | 15071 | 8.28 |
| -5 - 1.2 | High | 4996851 | 20.3 | 21844 | 12.00 | 0.5 - 0.7 | 3243717 | 13.2 | 61228 | 33.64 |
| 1.2 – 13 | Very high | 4982782 | 20.2 | 154863 | 85.09 | 0.7 - 1 | 2288737 | 9.3 | 100612 | 55.28 |

| FRFSI | Class | FRFSP | % FSM | VFP | % VF | Methods | Success Rate Curve, AUC % | | Prediction Rate Curve AUC % | |
|---|---|---|---|---|---|---|---|---|---|---|
| 4 – 9 | Very low | 3480969 | 14.1 | 15 | 0.01 | LR | 75.6 | | 81.4 | |
| 9 – 14 | Low | 5825312 | 23.6 | 511 | 0.28 | FR | 97.9 | | 99.1 | |
| 14 - 19 | Moderate | 5356987 | 21.7 | 4775 | 2.62 | | | | | |
| 19 - 27 | High | 4874089 | 19.8 | 21635 | 11.89 | IV | 71 | | 78.2 | |
| 27 - 46 | Very high | 5105544 | 20.7 | 155072 | 85.20 | | | | | |

| AHPFSI | Class | AHPFSP | %FSM | VFP | % VF | AHP | 82.5 | | 86.9 | |
|---|---|---|---|---|---|---|---|---|---|---|
| 0.5 - 1.7 | Very low | 4849344 | 19.7 | 0 | 0.00 | | | | | |
| 1.7 - 2.3 | Low | 6122024 | 24.8 | 12 | 0.01 | | | | | |
| 2.3 – 2.9 | Moderate | 6298368 | 25.6 | 558 | 0.31 | | | | | |
| 2.9 - 3.6 | High | 4887029 | 19.8 | 10491 | 5.76 | | | | | |
| 3,6 – 5.3 | Very high | 2486139 | 10.1 | 170947 | 93.92 | | | | | |

Note: AHPFSI is analytical hierarcky process flood susceptibility index, IVFSI is information value flood susceptibility index, IVFSP is information value flood susceptibility pixel, FSM is flood susceptibility map, VFP is validation flood pixel, VF is validation flood, LRFSI is logistic regression flood susceptibility index, LRFSP is logistic regression flood susceptibility pixel, FRFSI is frequency ratio flood susceptibility index, FRFSP is frequency ratio flood susceptibility pixel



## 4.2 Model Validation and Comparison

The most important ambition of flood susceptibility mapping is to determine the areas that are prone to flood hazards. However, flood susceptibility modeling without predication and model performance evaluation is non-sense to the application of disaster reduction programs. Although researchers used many techniques to validate the flood susceptibility model, the receiver operating characteristics (ROC) method is routinely used (Shafizadeh et al., 2018; Tehrany et al., 2013; Liuzzo et al., 2019) because of its simplify and produce clear as well as reliable results (Samanta et al., 2018; Rhmati et al., 2016; Khosravi et al., 2016; Pradhan and Lee, 2010). Therefore, the prediction and model performance of flood susceptibility map of the study area was validated by comparing the flood model with existing flood data using the ROC curve (Lee et al., 2007; Tien Bui et al., 2012; Pourghasemi et al., 2012). The prediction accuracy and model performance of the flood susceptibility map was evaluated quantitatively using the receiver operating characteristics (ROC) curve based on the evaluation of the true and false positive rates (Chauhan et al., 2010; Mahyat et al., 2019). Both the training and testing dataset were used to calculate the success rate curve and predictive rate curve. The predictive rate curve for the four models was obtained by comparing testing flood datasets with flood susceptibility index while the success rate curve also obtained for the four models by comparing training flood datasets. The AUC value ranges from 0.5 – 1 (Yesilnacar and Topal, 2005) and the closer the value to one indicating the higher accuracy of the model. As the results of the Success rate curve of AUC analysis indicated in (Table 4 and Fig. 9), FR has received a 97.9% and 99.1% success rate curve and prediction rate curve, respectively. When evaluating the accuracy of the model, the FR model indicated superior performance (97.9%), followed by the AHP model (82.5%), LR model (75.6%), and then the IV model (71%). Similarly, the model has the greatest prediction capacity (99.1%), followed by the AHP model (86.9%), LR model (81.4%) and the IV model has 78.2%. From the AUC results, the FR model indicating, the highest model accuracy and prediction capacity but the IV model has indicated relatively less model accuracy and predictive capacity in the present study. Moreover, the four models (FR, AHP, LR, and IV) resulted in AUC > 75% which is good, very good, and excellent model performance (Yesilnacar and Topal, 2005), respectively. This finding is similar to the work of (Bui et al., 2018; Samanta et al., 2018a; Rahman et al., 2019). Besides the ROC curve, flood-testing datasets that are not used for model development were overlaid on the four flood susceptible maps. The number of flood points that fells in the very high susceptibility class was



measured as shown in Table 4, 85.2%, 55.3%, 85.1% and 93.92% of flood points were fell in very
high susceptibility class of FR, LR, IV and AHP models. Here also the FR and AHP models
confirms again its excellent performance followed by the IV model. All in all the flood points
which fell in very high susceptibility class are greater than 55%, indicating acceptable model
accuracy of IV, LR, AHP and FR models.
Although the analytical hierarchy process, frequency ratio, information value, and logistic
regression methods are routinely used methods for flood susceptibility mapping, they have some
foreseeable limitations. For example, the logistic regression model can perform multivariate
statistical analysis between a dependent variable and a set of independent variables (Table 2), but
it is incapable to analyze the impacts of internal classes of flood governing factors individually on
flood occurrence. As the results indicated in Table 2, the importance of flood driving factors is
determined using the LR model. The result showed that among eleven factors, distance to stream
(1.73), elevation (0.8), slope gradient (0.769), flow accumulation (0.222), land use (0.159), soil
texture (0.106), and groundwater depth (0.006) had received the highest statistical impact on the
probability of flood occurrence (Table 2). These are in line with the finding of Kia et al., 2012;
Chapi et al., 2017; Mosavi et al., 2018; Falah et al., 2019; Rahman et al., 2019). Overall, logistic
regression also causes oversimplification and generalization on the effects of flood governing
factors. Whereas frequency ratio and information value are simple and effective statistical methods
that can extract the influence of each flood governing factor class on flood occurrence (Table 1),
but it cannot consider the relationship between these flood governing factors and flood occurrence.
The analytical hierarchy process method is very important methods to evaluate the effects of
factors and factor classes on flood occurrence probability, however, this method has a series of
subjectivity problem during pairwise comparison to assign the weights for each factor class and
flood driving factors. In summary, there is no unique statistical and expert based methods to
determine both the effects of each factor classes and general effects of flood factors. Therefore, a
combination use of bivariate and multivariate statistical methods to predict flood susceptibility in
a region is very essential when there is no a unique method that help to evaluate the effects of flood
driving factors as general and inherently.
In literature, comparison among information value, logistic regression, frequency ratio and
analytical hierarchy process method was not performed rather than the frequency ratio method



with the information value method, logistic regression method with information value and
frequency methods, the AHP method with the information value method, and the AHP method
with the frequency ratio method. (Chen et al., 2016) states that the prediction rate of 83.69% using
the frequency ratio model is better than the prediction rate of 81.22% using the information value
method. This finding is similar to the present study, the frequency ratio method showed better
performance for both success rates (AUC =97.9%) and predictive rate curve (AUC= 99.1%) than
the information value method with success rate curve (AUC = 71.0%) and predictive rate curve
(AUC = 78.2%).  As shown from the work of (Mahyat et al., 2018), the logistic regression model
showed a high predictive accuracy of AUC value of 79.45 % compared to the frequency ratio and
information value model with prediction rate curve value (AUC = 67.33% for FR, AUC=78.18%
for IV). Nevertheless, in the present model, the frequency model showed a relatively few
difference in prediction rate value (AUC = 99.1 %) than the information value and logistic
regression models with prediction rate value (AUC = 78.2% for IV, AUC=81.4% for LR). From
the work of (Khosravi et al., 2016), based on the predictive rate value of the area under the receiver
operating characteristic curve (AUC), the frequency ratio (FR) and analytical hierarchy process
(AHP) models showed a little bit different in predictive capacity, which is 96.57% for the FR
model and 94.92% for the AHP model. This result is in line with the present work, the prediction
rate of 99.1% using the frequency ratio model is better performance than the prediction rate curve
86.9% for the AHP model. Rahman et al., (2019) found that the logistic regression model
(AUC=86.8%) gave a more realistic flood susceptibility map than the frequency ratio (AUC=
85.6%) and AHP (AUC= 64%) model. However, this result is not in line with the present work
which is the frequency ratio is better than AHP and the logistic regression model. This difference
happens mostly due to the number of and types of input parameters for model construction.
Generally, the AHP, bivariate, and multivariate statistical methods in literature and this study
showed, the closer prediction capacity with AUC > 64% and AUC > 75%, respectively fell in the
range of good and very good/excellent performance (Yesilnacar and Topal, 2005). The flood
validation results for the four models (FR, LR, IV & AHP) are closer to each other. Therefore,
from these results, the research work finds out that in flood susceptibility mapping, the four models
have equal potential to generate flood-prone areas but factor selection should be playing a more
important role than the methods. Although all statistical models indicated higher prediction
accuracy, based on their statistical significance analysis result of AUC value (see Table ), the



frequency ratio (FR) model is better than the analytical hierarchy process (AHP), logistic
regression (LR) model, and information value model for regional land use planning, flood hazard
mitigation, and prevention purposes.

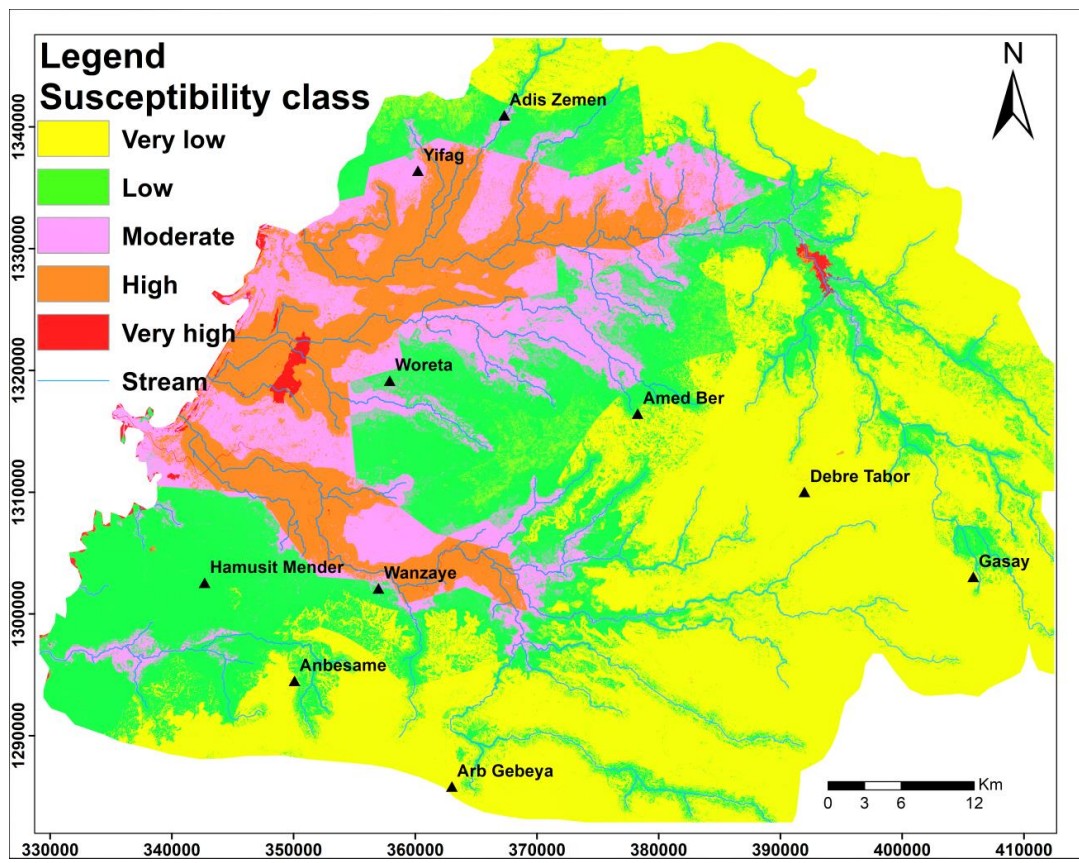

Figure 5 Flood Susceptibility map using frequency ratio method



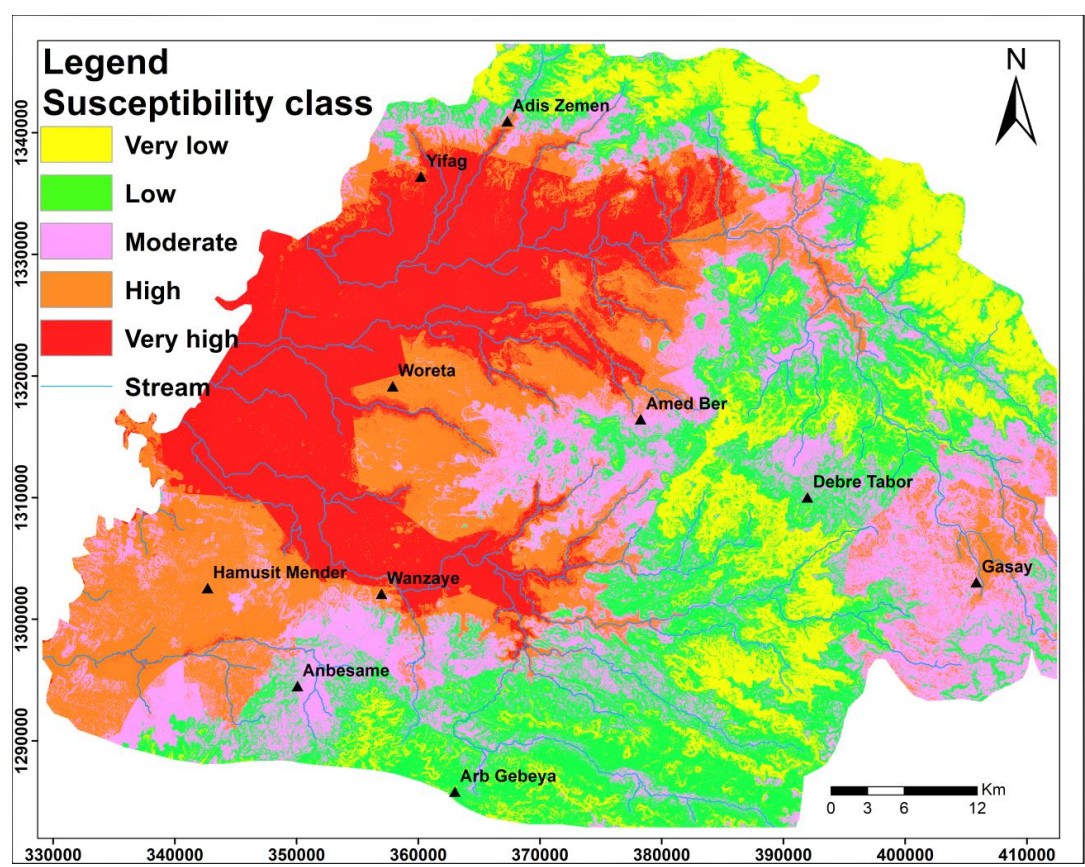

Figure 6 Flood Susceptibility map using information value method





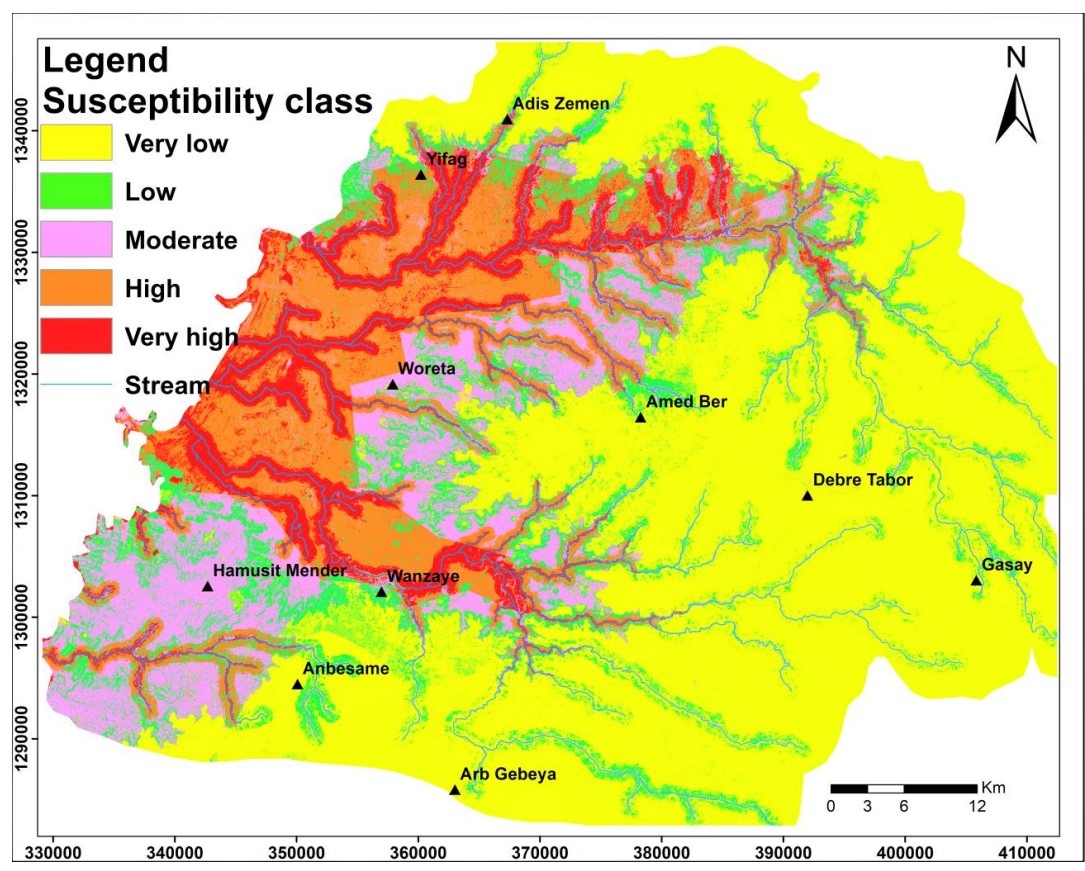

Figure 7 Flood Susceptibility map using logistic regression method


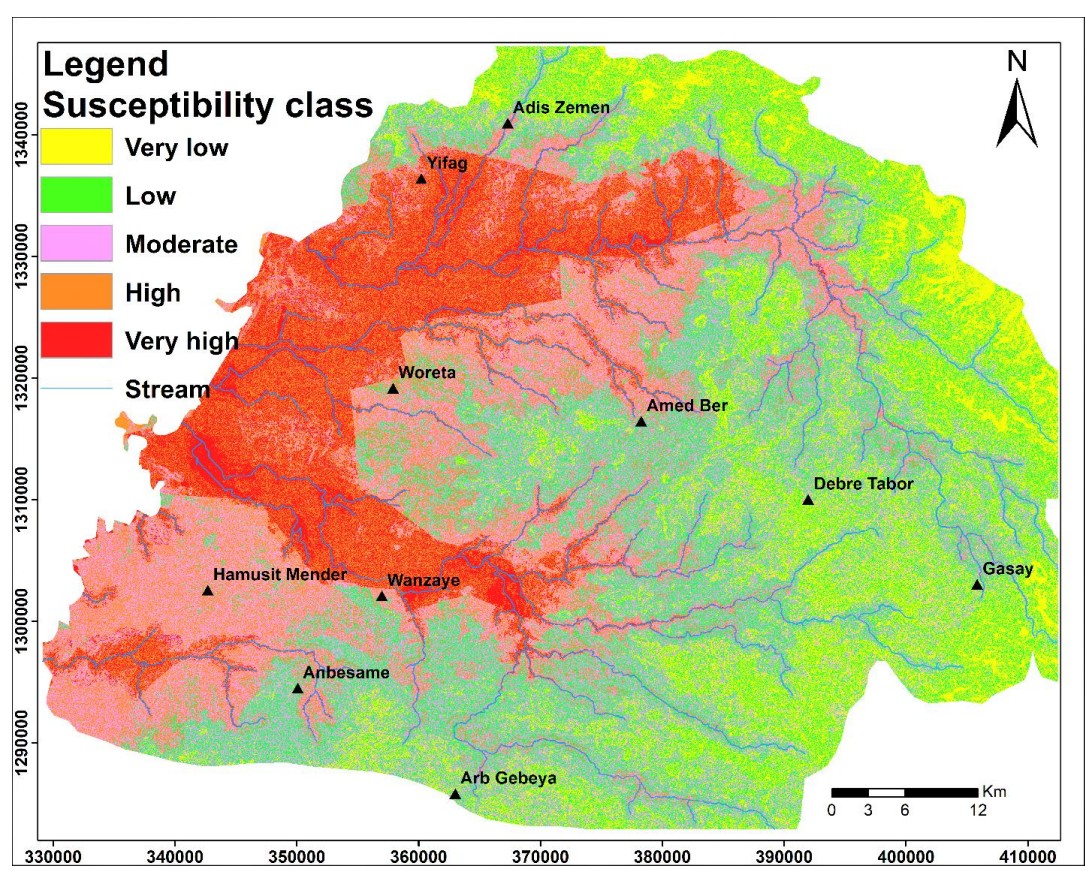


Figure 8 Flood Susceptibility map using analytical hierarchy process  method




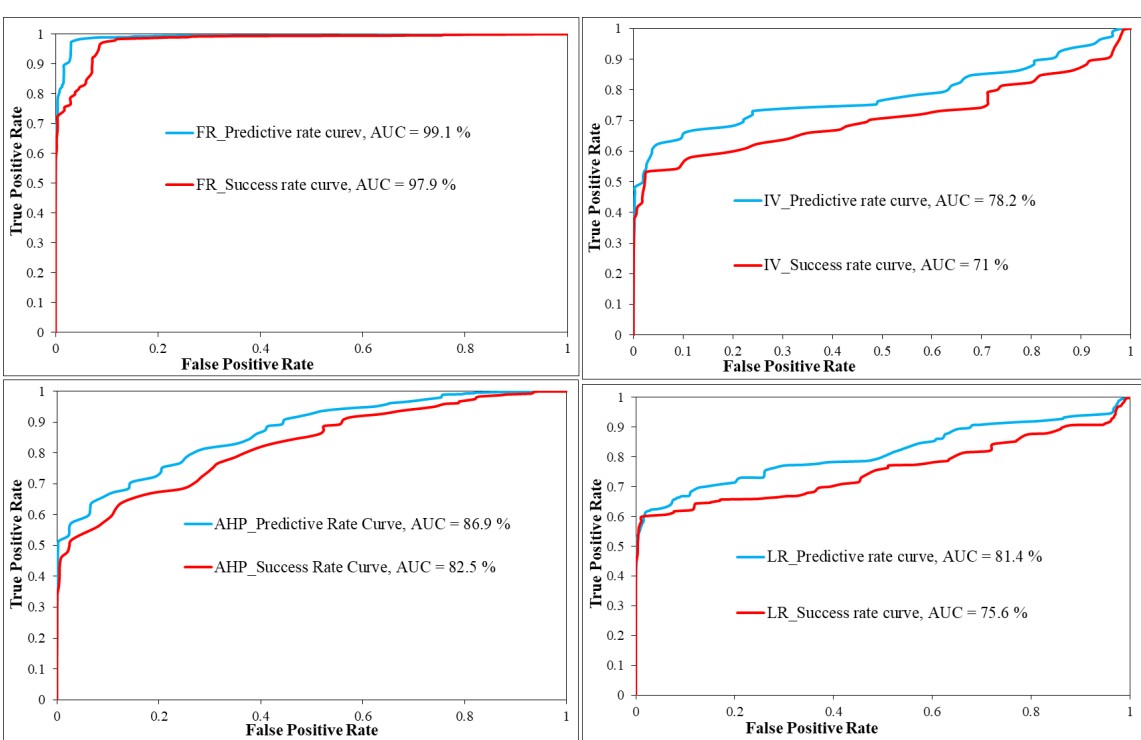

Figure 9 Predictive and success rate curves for IV, LR, FR and AHP methods

**Conclusion**
In flood hazard reduction and mitigation management, flood susceptibility map is one of the key element.
Therefore, it is essential to prepare the most precise and reliable flood susceptibility map. The application
of frequency ratio, information value, logistic regression, and analytical hierarchy process (AHP) models
have been tested in flood susceptibility mapping and their results are compared to each other using AUC
results. The results showed that the flood susceptibility map produced by the frequency ratio method is
relatively better than the AHP, logistic regression, and information value methods. However, the ranges of
prediction accuracy value for all four methods are indicated that the frequency ratio, AHP, logistic
regression, and information value methods are capable to produce an acceptable flood susceptibility model.
The models, which are generated using the bivariate, multivariate statistical, and AHP models, can
help to understand the flood hazard problems in the study area. Although the resulting maps cannot
forecast the time, and how often it can occur, it has provided the spatial distribution of flood
probability. These models can also provide important information to the researchers, local people,





government, and planners to reduce the flood hazard problems in the study area. Therefore, the
concerned bodies may at the Wereda/District, Zone, Region, and Federal levels take tangible
activities to mitigate the flood problem by avoiding permanent activities at the high and very high
regions with the integration of construction of check dams for streams.

**Author contributions**
Azemeraw Wubalem has conceptualized statistical analysis and done the completed modeling
analysis. Azemeraw Wubalem wrote the original drafts, which was reviewed and edited by all co-
authors. All authors have their contributions to writing the manuscript.
**Competing interest**
We declare that we do not any conflict of interest.
**Acknowledgments**
We would like to thank the University of Gondar for financial and equipment supports. We also
would like to thank all contributing project partners. We also want to give our special thanks to
the West Amhara Meteorological Agency, Amhara Water Well Drilling Enterprise, and Ethiopian
Mapping Agency.









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
