# Peer review of "Comparison of statistical and analytical hierarchy process methods on flood susceptibility"

_Natural Hazards and Earth System Sciences, 2020_

## Referee Comment (RC1) · Anonymous Referee #1 · 17 Dec 2020

Dear Editor and Authors,

the manuscript "Comparison of statistical and analytical hierarchy process methods on flood susceptibility mapping: in a case study of Tana sub-basin in northwestern Ethiopia" unfortunately does not present enough scientific significance, scientific quality and presentation quality at this stage.

The authors compared different methods for susceptibility mapping with the objective of choosing the best performing one for the region of lake Tana in Ethiopia. Although

the manuscript in principle would fit the scope of NHESS it is not relevant or, at best, is far from ready for publication in NHESS. This is not to say that the study was not a good technical exercise for the region of lake Tana in Ethiopia.

The manuscript does not address any relevant scientific or technical questions the way it is written. There is a widespread number of papers comparing the methods presented by the authors and it is hard to justify publishing one more of such papers. The only novelty one could perhaps find is the region analysed and the flood dataset used (never used before?). However, simply saying that data was collected from historical records and extensive fieldwork and giving no further detail is unacceptable.

Additionally, a number of strong statements are included in the manuscript without any proper demonstration or reference, for example "even though flood is one of the natural parts of the hydrological cycle, it is increased in both frequency and magnitude from year to year."

Finally, the use of English is poor and the authors give too much attention to describe well-established methods (e.g., logistic regression, GIS...) and terms (e.g., susceptibility, risk...), which make the manuscript very dense (and intractable). This work could be reported in a few pages, focusing on what is essential.

I recommend that, if the authors decide to resubmit the manuscript, please substantially revise the concept focusing on a stronger research question, the unquestionable novelty and the value that the manuscript brings to the scientific community. I also recommend the authors to be very concise and straight to the point, substantially improve the use of English, to be rigorous with the use of technical terms, and careful with statements that are not demonstrated/demonstrable.

―――――――――――――――――

---

## Referee Comment (RC2) · Guy J.-P. Schumann (Referee) · 12 Jan 2021

This paper uses a statistical method to derive a flood susceptibility map using a relatively large database of flood event points and a number of relevant flood "factor" maps.

The paper is in itself an interesting application, although not much new novelty is presented. Nonetheless, it is an interesting use case for Ethiopia.

This said, there are several major points that should be addressed in a much revised

version in my opinion. I highlighted those below.

- The English language should be revised;

- The statistical method used is sound and is appropriate for the task presented; however, the flood factor maps prepared as input are of course highly correlated with the flood event point locations chosen, so there is little surprise that this shows very good correlations, and, as a result, a very good output. In my opinion, the use of these factor maps needs to be better explained and in particular why their presented statistical method should be preferred over a simple topographic wetness index (TWI) for example, which would no doubt produce a very good flood susceptibility map too;

- In my opinion, the authors need to quantitatively demonstrate that a traditional, well-established GIS-type algorithm, such as TWI or other, is performing less well than the proposed statistical model. In other words, a TWI map could be taken as a benchmark of acceptability. This way, the authors could then also discuss more objectively the value of their proposed methods. In addition, it would allow them to use the entire flood event database as validation, instead of only 30% of it;

- Finally, I have some issue with the location of the selected flood events. Those are all located along the main river network, which makes it easy for any flood "factor" or indeed the proposed statistical method to perform well. Was there no historical flood event off the main river floodplains, in other words, was there no flood event that created flooding away from rivers? This would be interesting to assess, but at least should be explained.

---

## Author Comment (AC1) · 12 Jan 2021

Dear Reviewer, we appreciate your constructive comments and suggestions which helped us a lot in improving the quality of our manuscript. We do hope that your comments, suggestions, and concerns were addressed in our revised manuscript. Having said this, we will proceed to the responses to questions and comments. 1. The way of presentation in the manuscript is followed the scientific way of manuscript writing even though you could not agree. For example, this work is developed based on both secondary and primary input data. To use quality flood points as input for analysis, the

detailed flood points were collected using extensive fieldworks and Google Earth Imagery analysis. The manuscript consists of background, problem statement, objective, methods, results, discussion, and conclusions. 2. The comment that "this paper lacks scientific merit" is not convincing to us. Although the spatial relationship of the flood driving factors and flood points was performed using the existing information value, frequency ratio, analytical hierarchy process, and logistic regression methods, but still there is some contribution as the analysis and understanding will be different from one researcher to the other and with the existing area differences in different researchers who applied the same methods. This work has a great contribution to 1) add flood points in the regional and national geodatabases 2) will be used as a guideline for land-use planners, decision and policy makers 3) it may be used for regional flood mitigation purposes. Although flood susceptibility mapping is performed using both statistical and analytical hierarchy process methods in the globe, there are relevant differences in input parameters including flood factors and flood points, in the environmental condition of a region. Besides, there are no works were performed to compare the results of FR, LR, IV, and AHP in a single paper rather than they were applied individually. 3. As per your comments, several strong statements are included in the manuscript without any proper demonstration or reference, for example, "even though the flood is one of the natural parts of the hydrological cycle, it is increased in both frequency and magnitude from year to year" is acceptable it was type error we have corrected it for final revision. 3. As per your comments, we have substantially improved the manuscript of its grammatical, typological, and structural problems.

---

## Author Comment (AC2) · 12 Jan 2021

Dear Reviewer, we appreciate your constructive comments and suggestions which helped me a lot in improving the quality of my manuscript. We do hope that your comments, suggestions, and concerns were addressed in my revised manuscript. Having said this, we will proceed to the responses to questions and comments. 1. English language As per your comments, we have made substantial improvements on the grammatical, typological, and sentence structure problems throughout the manuscript because it also one of the comments by reviewer #1. 2. It is well known that TWI is

a commonly used very simple traditional flood hazard mapping, however, this method is used considering only the slope and flow accumulation in a region. Nevertheless, flood hazard is the result of the combination of several factors like soil texture, depth of groundwater, land use, vegetation cover, elevation, rainfall, stream density and distance from streams, the depth of riverbank. To solve the above-mentioned limitation of TWI, the use of statistical and analytical hierarchy process methods is very important to evaluate the spatial statistical correlation of flood driving factors and flood points. We have accepted your comments to explain the advantage and disadvantages of TWI in the introduction sections as per your comments. 3. It is possible to use all flood points for model development and validation, however, it resulted in a less reliable model and validation because it uses the same flood data layer for both training and validation purposes. Nevertheless, it can reduce sample size and sampling bias. The most important method is classified flood pixels as training and the testing data layer. 4. Because in the study area, floods frequently occur along the river floodplain, and backflush from Lake Tana that over flooded to certain areas. This is the reason why flood points are concentrated along riverbeds.